# AN UPPER BOUND FOR THE DISTRIBUTION OVERLAP INDEX AND ITS APPLICATIONS

## ABSTRACT

The overlap index between two probability distributions has various applications in statistics, machine learning, and other scientific research. However, approximating the overlap index is challenging when the probability distributions are unknown (i.e., distribution-free settings). This paper proposes an easy-to-compute upper bound for the overlap index without requiring any knowledge of the distribution models. We first utilize the bound to find the upper limit for the accuracy of a trained machine learning model when a domain shift exists. We additionally employ this bound to study the distribution membership classification of given samples. Specifically, we build a novel, distribution-free, computation-efficient, memory-efficient one-class classifier by converting the bound into a confidence score function. The proposed classifier does not need to train any parameters and is empirically accurate with only a small number of in-class samples. The classifier shows its efficacy and outperforms many state-of-the-art methods on various datasets in different one-class classification scenarios, including novelty detection, out-of-distribution detection, and backdoor detection. The obtained results show significant promise toward broadening the applications of overlap-based metrics.

## 1 INTRODUCTION

The distribution overlap index refers to the area intersected by two probability density functions (i.e., Fig. 1(a)) and measures the similarity between the two distributions. A high overlap index value implies a high similarity. Although the overlap index has various applications in many areas, such as biology (Langøy et al., 2012; Utne et al., 2012), economics (Milanovic & Yitzhaki, 2002), and statistics (Inman & Bradley Jr, 1989), the literature on approximating it under distribution-free settings is thin. This work proposes an upper bound for the overlap index with distribution-free settings to broaden the potential applications of overlap-based metrics. The bound is easy to compute and contains three terms: a constant number, the norm of the difference between the two distributions' means, and a variation distance between the two distributions over a subset. Even though finding such an upper bound for the distribution overlap index is already valuable, we further explore two additional applications of our bound as discussed below.

One application of our bound is for domain shift analysis. Specifically, a domain shift is a change in the dataset distribution between a model's training dataset and the testing dataset encountered during implementation (i.e., the overlap index value between the distributions of the two datasets is less than 1). We calculated the model's testing accuracy in terms of the overlap index between the distributions of the training and testing datasets and further found the upper limit of the accuracy using our bound for the overlap index. Knowing the upper bound for a model's testing accuracy helps measure the model's potential and compare it with other models. We validated the calculated upper limit accuracy with experiments in backdoor attacks.

Another application of our bound is for one-class classification. Specifically, one-class classification refers to a model that outputs positive for in-class samples and negative for out-class samples that are absent, poorly sampled, or not well defined (i.e., Fig. 1(b)). We propose a novel one-class classifier by converting our bound into a confidence score function to evaluate if a sample is in-class or out-class. The proposed classifier has many advantages. For example, implementing deep neural network-based classifiers requires training thousands of parameters and large memory, whereas implementing our classifier does not. It only needs sample norms to calculate the confi-

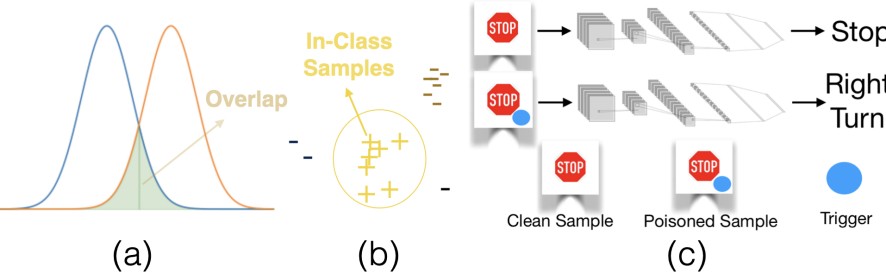

Figure 1: (a): Overlap of two distributions. (b): One-class classification. (c): Backdoor attack.

dence score. Besides, deep neural network-based classifiers need relatively large amounts of data to avoid under-fitting or over-fitting, whereas our method is empirically accurate with only a small number of in-class samples. Therefore, our classifier is computation-efficient, memory-efficient, and data-efficient. Additionally, compared with other traditional one-class classifiers, such as Gaussian distribution-based classifier, Mahalanobis distance-based classifier (Lee et al., 2018) and one-class support vector machine (Schölkopf et al., 2001), our classifier is distribution-free, explainable, and easy to understand.

Overall, the contributions of this paper include:

- Finding a distribution-free upper bound for the overlap index.
- Applying this bound to the domain shift analysis problem with experiments.
- Proposing a novel one-class classifier with the bound being the confidence score function.
- Evaluating the proposed one-class classifier through comparison with various state-of-the-art methods on several datasets, including UCI datasets, CIFAR-100, sub-ImageNet, etc., and in different one-class classification scenarios, such as novelty detection, out-of-distribution detection, and neural network backdoor detection.

## 1.1 BACKGROUND AND RELATED WORKS

**Measuring the similarity between distributions**: Gini & Livada (1943) and Weitzman (1970) introduced the concept of the distribution overlap index. Other measurements for the similarity between distributions include the total variation distance, Kullback-Leibler divergence (Kullback & Leibler, 1951), Bhattacharyya's distance (Bhattacharyya, 1943), and Hellinger distance (Hellinger, 1909). In psychology, some effect size measures' definitions involve the concept of the distribution overlap index, such as Cohen's U index (Cohen, 2013), McGraw and Wong's CL measure (McGraw & Wong, 1992), and Huberty's I degree of non-overlap index (Huberty & Lowman, 2000). However, they all have strong distribution assumptions (e.g., symmetry or unimodality) regarding the overlap index. Pastore & Calcagnì (2019) approximates the overlap index via kernel density estimators.

**One-class classification**: Moya & Hush (1996) coined the term one-class classification. One-class classification intersects with novelty detection, anomaly detection, out-of-distribution detection, and outlier detection. Yang et al. (2021) explains the differences among these detection areas. Khan & Madden (2014) discusses many traditional non neural network-based one-class classifiers, such as one-class support vector machine (Schölkopf et al., 2001), decision-tree (Comité et al., 1999), and one-class nearest neighbor (Tax, 2002). Two neural network-based one-class classifiers are (Ruff et al., 2018) and OCGAN (Perera et al., 2019). Morteza & Li (2022) introduces a Gaussian mixture-based energy measurement and compares it with several other score functions, including maximum softmax score (Hendrycks & Gimpel, 2017), maximum Mahalanobis distance (Lee et al., 2018), and energy score (Liu et al., 2020a) for one-class classification.

**Neural network backdoor attack and detection**: Gu et al. (2019) and Liu et al. (2018b) mentioned the concept of the neural network backdoor attack. The attack contains two steps: during training, the attacker injects triggers into the training dataset; during testing, the attacker leads the network to misclassify by presenting the triggers (i.e., Fig. 1(c)). The data poisoning attack (Biggio et al., 2012) and adversarial attack (Goodfellow et al., 2014) overlap with the backdoor attack. Some proposed

trigger types are Wanet (Nguyen & Tran, 2021), invisible sample-specific (Li et al., 2021), smooth (Zeng et al., 2021), and reflection (Liu et al., 2020b). Some methods protecting neural networks from backdoor attacks include neural cleanse (Wang et al., 2019), fine-pruning (Liu et al., 2018a), and STRIP (Gao et al., 2019). NNoculation (Veldanda et al., 2021) and RAID (Fu et al., 2022) utilize online samples to improve their detection methods. The backdoor detection problem also intersects with one-class classification. Therefore, some one-class classifiers can detect poisoned samples against the neural network backdoor attack.

**Organization of the Paper:** We first provide preliminaries and derive the proposed upper bound for the overlap index in Sec. 2. We next apply our bound to domain shift analysis in Sec. 3. We then propose, analyze, and evaluate our novel one-class classifier in Sec. 4. We finally conclude the paper in Section 5.

## 2 AN UPPER BOUND FOR THE OVERLAP INDEX

### 2.1 PRELIMINARIES

For simplicity, we consider the $\mathbb{R}^n$ space and continuous random variables. We also define $P$ and $Q$ as two probability distributions in $\mathbb{R}^n$ with $f_P$ and $f_Q$ being their probability density functions.

**Definition 1** (Overlap Index). *The overlap* $\eta : \mathbb{R}^n \times \mathbb{R}^n \to [0, 1]$ *of the two distributions is defined:*

$$\eta(P, Q) = \int_{\mathbb{R}^n} \min[f_P(x), f_Q(x)]dx. \tag{1}$$

**Definition 2** (Total Variation Distance). *The total variation distance* $\delta : \mathbb{R}^n \times \mathbb{R}^n \to [0, 1]$ *of the two distributions is defined as*

$$\delta(P, Q) = \frac{1}{2} \int_{\mathbb{R}^n} |f_P(x) - f_Q(x)| \, dx. \tag{2}$$

**Definition 3** (Variation Distance on Subsets). *Given a subset $A$ from $\mathbb{R}^n$, we define* $\delta_A : \mathbb{R}^n \times \mathbb{R}^n \to [0, 1]$ *to be the variation distance of the two distributions on $A$, which is*

$$\delta_A(P, Q) = \frac{1}{2} \int_A |f_P(x) - f_Q(x)|dx. \tag{3}$$

**Remark 1.** *One can prove that $\eta$ and $\delta$ satisfy the following equation:*

$$\eta(P, Q) = 1 - \delta(P, Q) = 1 - \delta_A(P, Q) - \delta_{\mathbb{R}^n \setminus A}(P, Q). \tag{4}$$

*The quantity $\delta_A$ defined in (3) will play an important role in deriving our upper bound for $\eta$.*

### 2.2 THE UPPER BOUND FOR THE OVERLAP INDEX

We now proceed with deriving our proposed upper bound.

**Theorem 1.** *Without loss of generality, assume $D^+$ and $D^-$ are two probability distributions on a bounded domain $B \subset \mathbb{R}^n$ with defined norm $|| \cdot ||$ [1] (i.e., $\sup_{x \in B} ||x|| < +\infty$), then for any subset $A \subset B$ with its complementary set $A^c = B \setminus A$, we have*

$$\eta(D^+, D^-) \leq 1 - \frac{1}{2r_{A^c}} ||\mu_{D^+} - \mu_{D^-}|| - \frac{r_{A^c} - r_A}{r_{A^c}} \delta_A \tag{5}$$

*where $r_A = \sup_{x \in A} ||x||$ and $r_{A^c} = \sup_{x \in A^c} ||x||$, $\mu_{D^+}$ and $\mu_{D^-}$ are the means of $D^+$ and $D^-$, and $\delta_A$ is the variation distance on set $A$ defined in **Definition** 3. Moreover, let $r_B = \sup_{x \in B} ||x||$, then we have*

$$\eta(D^+, D^-) \leq 1 - \frac{1}{2r_B} ||\mu_{D^+} - \mu_{D^-}|| - \frac{r_B - r_A}{r_B} \delta_A. \tag{6}$$

*Since (6) holds for any $A$, a tighter bound can be written as*

$$\eta(D^+, D^-) \leq 1 - \frac{1}{2r_B} ||\mu_{D^+} - \mu_{D^-}|| - \max_A \frac{r_B - r_A}{r_B} \delta_A. \tag{7}$$

---

[1] In this paper, we use the $L_2$ norm. However, the choice of the norm is not unique and the analysis can be carried out using other norms as well.

*Proof.* Let $f_{D^+}$ and $f_{D^-}$ be the probability density functions for $D^+$ and $D^-$. From (4), we have

$$\eta(D^+, D^-) = 1 - \delta_A(D^+, D^-) - \delta_{A^c}(D^+, D^-). \tag{8}$$

Using (8), triangular inequality, and boundedness, we obtain

$$||\mu_{D^+} - \mu_{D^-}|| = ||\int_B x \left(f_{D^+}(x) - f_{D^-}(x)\right) dx|| \leq \int_B ||x(f_{D^+}(x) - f_{D^-}(x))||dx \tag{9}$$

$$= \int_A ||x|| \cdot |f_{D^+}(x) - f_{D^-}(x)|dx + \int_{A^c} ||x|| \cdot |f_{D^+}(x) - f_{D^-}(x)|dx \tag{10}$$

$$\leq 2r_A\delta_A + 2r_{A^c}\delta_{A^c} = 2r_A\delta_A + 2r_{A^c}(1 - \delta_A - \eta(D^+, D^-)) \tag{11}$$

which implies (5). Replacing $r_{A^c}$ with $r_B$ in (11) implies (6). □

**Remark 2.** *The only assumption in this theorem is that the probability distribution domain is bounded. However, almost all real-world applications satisfy the boundedness assumption since the data is bounded. Therefore, $r_B$ can always be found (or at least a reasonable approximation can be found). Additionally, we can constrain $A$ to be a bounded ball so that $r_A$ is also known. Although the proof of this theorem involves probability density functions, the computation does not require knowing the probability density functions but only finite samples because we can use the law of large numbers to estimate $||\mu_{D^+} - \mu_{D^-}||$ and $\delta_A$, which will be shown next.*

## 2.3 APPROXIMATING THE BOUND WITH FINITE SAMPLES

Let $g : B \to \{0, 1\}$ be a condition function[2] and define $A = \{x \mid g(x) = 1, x \in B\}$. According to the definition of $\delta_A$ and triangular inequality, we have

$$\delta_A(D^+, D^-) = \frac{1}{2} \int_A |f_{D^+}(x) - f_{D^-}(x)|dx \geq \frac{1}{2}|\int_A f_{D^+}(x) - f_{D^-}(x)dx| \tag{12}$$

$$= \frac{1}{2} \left| \int_B f_{D^+}(x)g(x)dx - \int_B f_{D^-}(x)g(x)dx \right| = \frac{1}{2} |\mathbb{E}_{D^+}[g] - \mathbb{E}_{D^-}[g]|. \tag{13}$$

Calculating $\mathbb{E}_{D^+}[g]$ and $\mathbb{E}_{D^-}[g]$ is easy: one just needs to draw samples from $D^+$ and $D^-$, and then average their $g$ values. Applying (13) into **Theorem** 1 gives the following corollary:

**Corollary 1.** *Given $D^+$, $D^-$, $B$, and $|| \cdot ||$ used in **Theorem** 1, let $A(g) = \{x \mid g(x) = 1, x \in B\}$ with any condition function $g : B \to \{0, 1\}$. Then, an upper bound for $\eta(D^+, D^-)$ that can be obtained by our approximation is*

$$\eta(D^+, D^-) \leq 1 - \frac{1}{2r_B}||\mu_{D^+} - \mu_{D^-}|| - \max_g \frac{r_B - r_{A(g)}}{2r_B} |\mathbb{E}_{D^+}[g] - \mathbb{E}_{D^-}[g]|. \tag{14}$$

Given several condition functions $\{g_j\}_{j=1}^k$ and finite sample sets (i.e., $\{x_i^+\}_{i=1}^n \sim D^+$ and $\{x_i^-\}_{i=1}^m \sim D^-$), Alg. 1 shows how to compute the RHS of (14).

---

**Algorithm 1** ComputeBound($\{x_i^+\}_{i=1}^n$, $\{x_i^-\}_{i=1}^m$, $\{g_j\}_{j=1}^k$)

---

$B \leftarrow \{x_1^+, x_2^+, ..., x_n^+, x_1^-, x_2^-, ..., x_m^-\}$ and $r_B \leftarrow \max_{x \in B} ||x||$
$\Delta_\mu \leftarrow ||\frac{1}{n}\sum_{i=1}^n x_i^+ - \frac{1}{m}\sum_{i=1}^m x_i^-||$
**for** $j = 1 \to k$ **do**
    $A = \{x \mid g_j(x) = 1, x \in B\}$ and $r_A \leftarrow \max_{x \in A} ||x||$
    $s_j \leftarrow \left(1 - \frac{r_A}{r_B}\right) \left|\frac{1}{n}\sum_{i=1}^n g_j(x_i^+) - \frac{1}{m}\sum_{i=1}^m g_j(x_i^-)\right|$
**end for**
**Return:** $1 - \frac{1}{2r_B}\Delta_\mu - \frac{1}{2}\max_j s_j$

---

**Remark 3.** *The choice of condition functions is not unique. In this work, we use the indicator function $g(x) = \mathbb{1}\{||x|| \leq r\}$, which outputs 1 if $||x|| \leq r$ and 0 otherwise. By setting different values for $r$, we generate a family of condition functions. The motivation for choosing such indicator function form is that it is the most simple way to separate a space nonlinearly and it saves computations by directly applying $r$ into **Corollary 1**. However, other indicator functions, such as RBF kernel-based indicator functions, are worth exploring and will be considered in our future works.*

---

[2]The condition function is an indicator function $\mathbb{1}\{condition\}$ that outputs 1 when the input satisfies the given condition and 0 otherwise.

## 3 APPLICATION OF OUR BOUND TO DOMAIN SHIFT ANALYSIS

We now apply our bound to domain shift analysis.

**Theorem 2.** *Assume that $D$ and $D^*$ are two different data distributions (i.e., $\eta(D, D^*) < 1$). If a model is trained on $D$ with $p$ accuracy on $D$ and $q$ accuracy on $D^* \setminus D$, then the overall accuracy of the model on $D^*$ is $p\eta(D, D^*) + q(1 - \eta(D, D^*))$, which is upper bounded because $\eta(D, D^*)$ is upper bounded by (14).*

**Remark 4.** *To prove the theorem, let $f_D$ and $f_{D^*}$ be their probability density functions, then*

$$Accuracy = \int_{x \sim D^*} \left( p\frac{\min\{f_D(x), f_{D^*}(x)\}}{f_{D^*}(x)} + q\left(1 - \frac{\min\{f_D(x), f_{D^*}(x)\}}{f_{D^*}(x)}\right) \right) f_{D^*}(x)dx \quad (15)$$

$$= p\eta(D, D^*) + q(1 - \eta(D, D^*)) = (p - q)\eta(D, D^*) + q. \quad (16)$$

*Without loss of generality, assume $p > q$. Then (16) shows that a large domain shift (i.e., a small $\eta(D, D^*)$) leads to low overall accuracy of the model on the testing data distribution $D^*$. If $p = 1$ and $q = 0$, then the overall accuracy is equal to $\eta(D, D^*)$.*

**Theorem 2 in Backdoor Attack Scenarios:** A backdoor attack scenario (Fig. 1(b)) considers that the model has a zero accuracy on poisoned data distribution as the attack success rate is almost 100%. Define the clean data distribution as $D$, poisoned data distribution as $D^p$, and a testing data distribution $D^*$ composed by $D$ and $D^p$, i.e., $D^* = \sigma D + (1 - \sigma)D^p$, where $\sigma \in [0, 1]$ is the purity ratio (i.e., the ratio of clean samples to the entire testing samples). With all the settings above, we know that $q = 0$ on $D^p$ and (16) becomes

$$Accuracy = p\eta(D, D^*) \leq p(1 - \frac{1}{2r_B}||\mu_D - \mu_{D^*}|| - \max_g \frac{r_B - r_{A(g)}}{2r_B}|\mathbb{E}_D[g] - \mathbb{E}_{D^*}[g]|) \quad (17)$$

$$= p(1 - \frac{1 - \sigma}{2r_B}||\mu_D - \mu_{D^p}|| - (1 - \sigma)\max_g \frac{r_B - r_{A(g)}}{2r_B}|\mathbb{E}_D[g] - \mathbb{E}_{D^p}[g]|). \quad (18)$$

(17) shows that the actual model accuracy on the contaminated testing data should be bounded by the multiplication of its accuracy $p$ on clean data and our upper bound for $\eta(D, D^*)$ with samples. (18) shows that the upper limit of the model accuracy on the contaminating data should linearly increase with the purity ratio $\sigma$ (i.e., the percentage of clean samples over the entire testing samples).

**Validating Theorem 2 in Backdoor Attack Scenarios**: The considered datasets are MNIST (LeCun et al., 2010), GTSRB (Stallkamp et al., 2011), YouTube Face (Wolf et al., 2011), and sub-ImageNet (Deng et al., 2009). We composed the testing datasets $D^*$ with $\sigma = 0, 0.1, ..., 0.9, 1$ and calculated the upper bound for $\eta(D, D^*)$ using $L_1$, $L_2$, and $L_\infty$ norms in the raw data space, model output space, and intermediate layer space. The actual model accuracy and corresponding upper bounds are plotted in Fig. 2. The actual model accuracy is below all the calculated upper limits, validating **Theorem** 2. Additionally, the upper limits grow linearly with $\sigma$, supporting (18). In other scenarios except for backdoor attacks, the model accuracy, $p$, and $q$ may be known, then the lower bound for $\eta$ can be estimated by **Theorem** 2. **Theorem** 2 can also help in finding $q$ by knowing $p$, $\eta$, and model accuracy. Therefore, **Theorem** 2 has practical relevance and usefulness.

## 4 APPLICATION OF OUR BOUND TO ONE-CLASS CLASSIFICATION

### 4.1 PROBLEM FORMULATION FOR ONE-CLASS CLASSIFICATION

Given $\mathbb{R}^d$ space and $n$ samples $\{x_i\}_{i=1}^n$ that lie in an unknown probability distribution, we would like to build a test $\Psi : \mathbb{R}^d \to \{\pm 1\}$ so that for any new input $x$, $\Psi(x)$ outputs 1 when $x$ is from the same unknown probability distribution, and outputs -1, otherwise. Some applications of $\Psi$ are novelty detection, out-of-distribution detection, and backdoor detection (e.g., Fig. 1(b, c)).

### 4.2 A NOVEL CONFIDENCE SCORE FUNCTION

Given some in-class samples $\{x_i\}_{i=1}^n$, one can pick several condition functions $\{g_j\}_{j=1}^k$, where $g_j(x) = \mathbb{1}\{||x|| \leq r_j\}$ for different $r_j$, so that $f(x) = ComputeBound(\{x\}, \{x_i\}_{i=1}^n, \{g_j\}_{j=1}^k)$ defined in Alg. 1 is a score function that measures the likelihood of any input, $x$, being an in-class sample. Alg. 2 shows the overall one-class classification algorithm. $k = 10$ in our experiments.

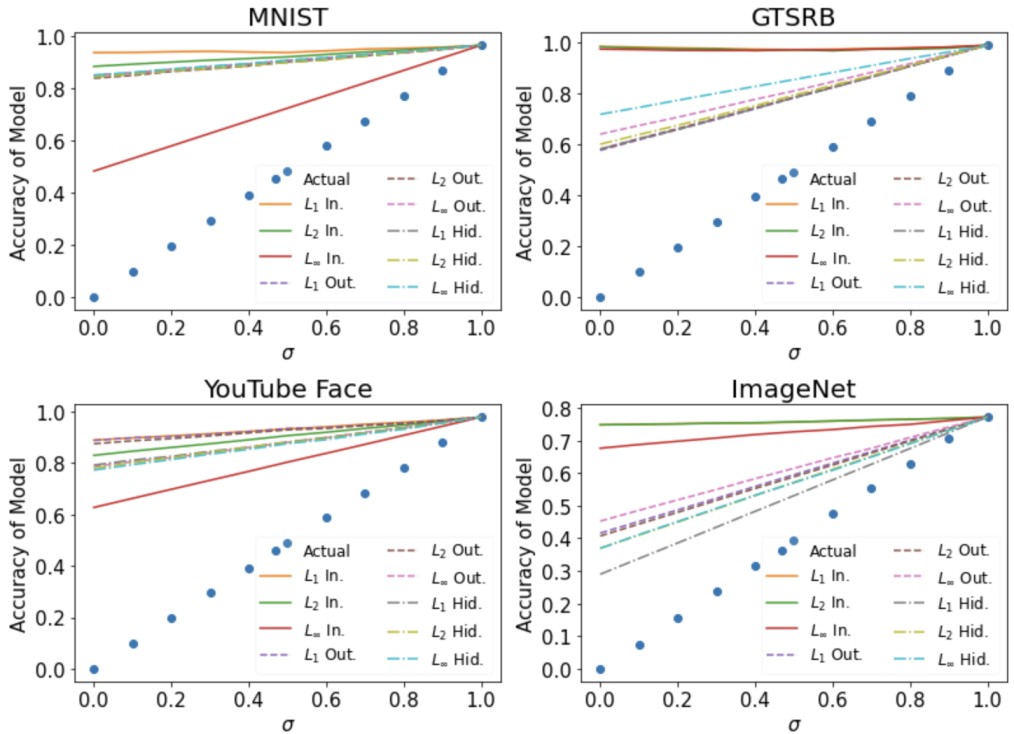

Figure 2: The actual model accuracy (dot) vs. (16) (solid) calculated with $L_1$, $L_2$, and $L_\infty$ norms in input, output, and hidden spaces. x: the ratio of clean samples to the entire testing samples.

---

**Algorithm 2** The Novel One-Class Classifier for the Input $x$

---

Given in-class samples $\{x_i\}_{i=1}^n$, select several condition functions $\{g_j\}_{j=1}^k$, set a threshold $T_0$
**if** $ComputeBound(\{x\}, \{x_i\}_{i=1}^n, \{g_j\}_{j=1}^k) \geq T_0$ **then**
    $x$ is an in-class sample
**else**
    $x$ is an out-class sample
**end if**

---

**Remark 5.** *The score function $f$ measures the maximum similarity between the new input $x$ and the available in-class samples $\{x_i\}_{i=1}^n$. Different $T_0$ lead to different detection accuracy. However, we will show that the proposed one-class classifier has an overall high accuracy under different $T_0$.*

### 4.3 COMPUTATION AND SPACE COMPLEXITIES

Our algorithm can pre-compute and store $\frac{1}{n}\sum_{i=1}^n x_i$ and $\frac{1}{n}\sum_{i=1}^n g_j(x_i)$ with $j = 1, 2, ..., k$. There-fore, the total space complexity is $\mathcal{O}(k+1)$. Assume that the total number of new online inputs is $l$; then, for every new input $x$, our algorithm needs to calculate $||x||$ once and $s_j$ for $k$ times. Therefore, the total computation complexity is $\mathcal{O}(l(k+1))$. Empirically, we restricted $k$ to be a small number (e.g., 10) so that even devices without strong computation power can run our algorithm efficiently. Therefore, our classifier is computation-efficient and memory-efficient.

### 4.4 EVALUATION

**Overall Setup:** All the baseline algorithms with optimal hyperparameters, related datasets, and models were acquired from the corresponding authors' websites. The only exception is backdoor detection, in which we created our own models. However, we have carefully fine-tuned the baseline

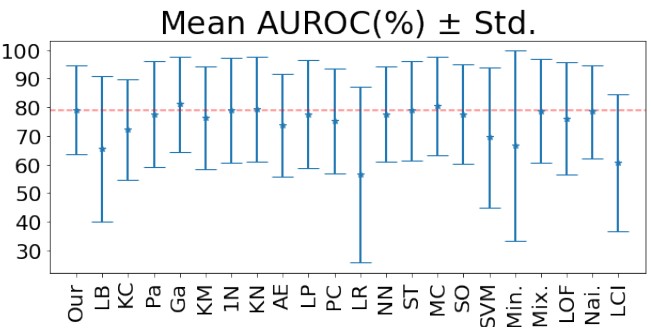

Figure 3: The method is listed in the same order as in Table 4.

methods' hyperparameters to ensure their best performance over other hyperparameter choices. Our approach used ten indicator functions for all the experiments.

### 4.4.1 ONE-CLASS CLASSIFICATION FOR NOVELTY DETECTION

We evaluated our classifier on 100 small UCI datasets (UCI; Dua & Graff, 2017) and recorded the area under the receiver operating characteristic curve (AUROC). Fig. 3 shows the mean and standard deviation of AUROC for ours and other classifiers. Detailed numerical results can be found in Table 4 in Appendix A. The implementation code is provided in the supplementary material.

Our classifier is the most consistent with the smallest standard deviation. Except for *Gaussian*, *K-Nearest Neighbor*, and *Minimum Covariance Determinant*, our classifier outperforms the other methods by showing the highest AUROC mean and the lowest AUROC standard deviation. Among *Gaussian*, *K-Nearest Neighbor*, and *Minimum Covariance Determinant*, *Gaussian* is the best classifier by having the highest mean and lowest standard deviation of AUROC. However, our method is comparable to *Gaussian* by showing a close mean and a smaller standard deviation.

Besides the results, our classifier is distribution-free, computation-efficient, and memory-efficient, whereas some other classifiers do not. Additionally, our method is also easy to explain and understand: the score measures the maximum similarity between the new input and the available in-class samples. Therefore, we conclude that our classifier is valid for novelty detection.

### 4.4.2 ONE-CLASS CLASSIFICATION FOR OUT-OF-DISTRIBUTION DETECTION

We used CIFAR-10 and CIFAR-100 testing datasets (Krizhevsky et al., 2009) as the in-distribution datasets. The compared methods contain MSP (Hendrycks & Gimpel, 2017), Mahalanobis (Lee et al., 2018), Energy score (Liu et al., 2020a), and GEM (Morteza & Li, 2022). We used WideResNet (Zagoruyko & Komodakis, 2016) to extract features from the raw data. The WideResNet models (well-trained on CIFAR-10 and CIFAR-100 training datasets) and corresponding feature extractors were acquired from Morteza & Li (2022). All the methods were evaluated in the same feature spaces with their optimal hyperparameters for fair comparisons. To fit the score function's parameters for all the methods, we formed a small dataset by randomly selecting 10 samples from each class. The out-of-distribution datasets include Textures (Cimpoi et al., 2014), SVHN (Netzer et al., 2011), LSUN-Crop (Yu et al., 2015), LSUN-Resize (Yu et al., 2015), and iSUN (Xu et al., 2015). We used three metrics: the detection accuracy for out-of-distribution samples when the detection accuracy for in-distribution samples is $95\%$ (TPR95), AUROC, and area under precision and recall (AUPR).

Table 1 shows the average results for CIFAR-10 and CIFAR-100. The details for each individual out-of-distribution dataset can be found in Table 5 and Table 6 in Appendix B. Our method outperforms the other methods by using the least memory and showing the highest AUROC on average. Our approach is also one of the fastest methods: the execution time of our approach for each sample is less than one millisecond (ms). For the CIFAR-10 case, our method also shows the highest average TPR95, and the average AUPR of our method is over 92%. For the CIFAR-100 case, the average TPR95 of our method is $0.3\%$ close to the highest average TPR95, and the average AUPR of our method is over 85%. For each individual out-of-distribution dataset, our method always outperforms no less than half of the methods in TPR95 and AUROC, and the total average AUPR of our method

Table 1: Average performance on various out-of-distribution datasets. Our method can be further improved with an iterative approach as shown in Table 2.

| In-Distributions | Method | TPR95 | AUROC | AUPR | Time/Sample | Memory |
|---|---|---|---|---|---|---|
| CIFAR-10 | Ours | **72.33%** | **92.76%** | 92.32% | 0.60ms | **1048.22MiB** |
| | MSP | 50.63% | 91.46% | **98.07%** | **0.02ms** | 1825.21MiB |
| | Mahala. | 46.83% | 90.46% | 97.92% | 30.61ms | 1983.17MiB |
| | Energy | 68.31% | 92.32% | 97.96% | 0.22ms | 1830.01MiB |
| | GEM | 50.81% | 90.45% | 97.91% | 25.62ms | 1983.51MiB |
| CIFAR-100 | Ours | 48.08% | **87.63%** | 85.72% | 0.60ms | **1134.32MiB** |
| | MSP | 19.87% | 75.97% | 94.09% | **0.02ms** | 1825.98MiB |
| | Mahala. | 48.35% | 84.90% | 96.37% | 56.24ms | 1983.81MiB |
| | Energy | 27.91% | 80.44% | 95.15% | 0.21ms | 1838.23MiB |
| | GEM | **48.36%** | 84.94% | **96.38%** | 56.27ms | 1984.77MiB |

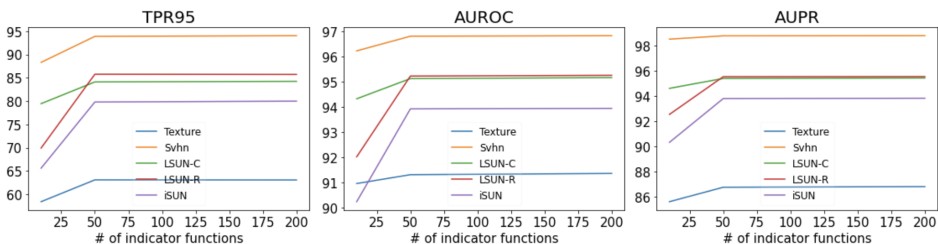

Figure 4: Improvements with more indicator functions with CIFAR-10 being the in-distribution data.

over all cases is 89.02%. Our method can be further improved with an iterative approach as shown in Table 2 with details discussed at the end. We noticed that the out-of-distribution datasets are much smaller in size than the in-distribution datasets. Therefore, although the current AUPR is sufficient to ensure that our approach is valid, we see a potential improvement in our method to increase the AUPR for heavily imbalanced problems in our prospective work. On balanced datasets, our approach shows higher AUPR than the baseline methods as shown in Table 3 with details in Table 7 in Appendix C. Therefore, our approach performs better on balanced datasets than on unbalanced datasets. We also empirically observed that the compared baseline methods reported errors when data dimensions are dependent because the compared baseline methods need to calculate the inverse of the estimated covariance matrices that will not be full rank if data dimensions are dependent. We have reported this observation in Table 7 in the appendix. In contrast, our approach works since it does not require finding the inverse of any matrices. Further, Table 1 and Table 3 together show that the baseline methods perform well only for out-of-distribution detection, whereas our approach performs well for both out-of-distribution detection and backdoor detection (details are explained in next subsection). In summary, our classifier is a valid out-of-distribution detector.

**Improvement with more indicator functions**: Our approach used ten indicator functions in all experiments. However, we have evaluated our approach by using more indicator functions (i.e., more $r_i$) and plotted the results in Fig. 4. From the figure, the performance of our approach increases with more indicator functions being used and eventually converges to a limit. This limit is determined by the out-of-distribution dataset, the tightness of our bound in **Corollary** 1, and the form of utilized indicator functions (i.e., $g(x) = \mathbb{1}\{||x|| \leq r\}$ in this work). To increase this limit on a given out-of-distribution dataset with the current form of our upper bound, a more advanced type of indicator function is required, which will be our future work as mentioned in **Remark** 3.

**Improvement with an iterative approach**: Except for using more indicator functions, our approach can also be improved by an interative approach with only the original ten indicator functions being used. Assume that the confidence score of an input $x$ is $s(x)$, then its iterative confidence score, $s'(x)$, can be calculated by Alg. 1 with the condition function $g(x) = \mathbb{1}\{s(x) \leq T_i\}$, where $T_i$ represents different thresholds. Table 2 shows the performance of our approach by applying the iterative score $s'(x)$ to Alg. 2. The results show considerable improvement compared to Table 1 for

Table 2: Performance of our approach with the iterative approach.

| In-Distributions | Metrics (%) | Texture | SVHN | LSUN-C | LSUN-R | iSUN | Ave. |
|---|---|---|---|---|---|---|---|
| CIFAR-10 | TPR95 | 64.20 | 94.10 | 83.63 | 85.41 | 79.62 | 81.39 |
| | AUROC | 92.00 | 96.88 | 95.29 | 95.48 | 94.46 | 94.82 |
| | AUPR | 92.02 | 98.78 | 95.45 | 95.65 | 94.65 | 95.31 |
| CIFAR-100 | TPR95 | 42.5 | 93.75 | 57.76 | 88.49 | 82.15 | 72.93 |
| | AUROC | 85.21 | 96.61 | 89.23 | 96.09 | 94.72 | 92.37 |
| | AUPR | 85.22 | 98.69 | 89.70 | 96.19 | 94.99 | 92.95 |

Table 3: Average performance for backdoor detection over various backdoor triggers and datasets.

| Metrics (%) | Ours | STRIP | Mahalanobis | GEM | MSP |
|---|---|---|---|---|---|
| TPR95 | 89.40 | 39.60 | 56.97 | **91.57** | 39.24 |
| AUROC | **96.68** | 70.30 | 75.94 | 58.08 | 54.92 |
| AUPR | **95.42** | 68.76 | 76.37 | 75.88 | 60.52 |

the average and to Table 5 and Table 6 for each individual out-of-distribution dataset. Exploring the potential improvement of our approach with more rounds of iterations will also be our future work.

### 4.4.3 ONE-CLASS CLASSIFICATION FOR BACKDOOR DETECTION

The utilized datasets are MNIST (LeCun et al., 2010), CIFAR-10 (Krizhevsky et al., 2009), GTSRB (Stallkamp et al., 2011), YouTube Face (Wolf et al., 2011), and sub-ImageNet (Deng et al., 2009). The adopted backdoor attacks include naive triggers, all-label attacks (Gu et al., 2019), moving triggers (Fu et al., 2020), Wanet (Nguyen & Tran, 2021), combination attacks, large-sized triggers, filter triggers, and invisible sample-specific triggers (Li et al., 2021), as listed in Fig. 5 in Appendix C. The neural network architecture includes Network in Network (Lin et al., 2014), Resnet (He et al., 2016), and other networks from (Wang et al., 2019; Gu et al., 2019). For each backdoor attack, we assume that a small clean validation dataset is available (i.e., 10 samples from each class) at the beginning. Therefore, the poisoned samples (i.e., samples attached with triggers) can be considered out-class samples, whereas the clean samples can be considered in-class samples. We used the backdoored network to extract data features. Then, we evaluated our one-class classifier and compared it with the previous baseline methods and STRIP (Gao et al., 2019) in the feature space. The metrics used are the same: TPR95 (i.e., the detection accuracy for poisoned samples when the detection accuracy for clean samples is 95%), AUROC, and AUPR. Table 3 shows the average performance. Details on each individual trigger can be found in Table 7 in Appendix C

From the table, our classifier outperforms other baseline methods on average by showing higher AUROC, and AUPR. As for TPR95, our approach is very close to GEM. Compared with STRIP on the overall average performance, our classifier is 49.8% higher in TPR95, 26.38% higher in AUROC, and 26.66% higher in AUPR. For each individual trigger, the TPR95 of our method is over 96% for most cases, the AUROC of our method is over 97% for most cases, and the AUPR of our method is over 95% for most cases. It is also seen that our classifier is robust against the latest or advanced backdoor attacks, such as Wanet, invisible trigger, all label attack, and filter attack, whereas the baseline methods show low performance on those attacks. Therefore, we conclude that our classifier is valid for backdoor detection.

## 5 CONCLUSION

This paper proposes an easy-to-compute distribution-free upper bound for the distribution overlap index. Two applications of the bound are explored. The first application is for domain shift analysis with a proposed theorem and discussion. The second application is for one-class classification. Specifically, this paper introduces a novel distribution-free one-class classifier with the bound being its confidence score function. The classifier is sample-efficient, computation-efficient, and memory-efficient. The proposed classifier is evaluated on novelty detection, out-of-distribution detection, and backdoor detection on various datasets and compared with many state-of-the-art methods. The obtained results show significant promise toward broadening the application of overlap-based metrics.

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

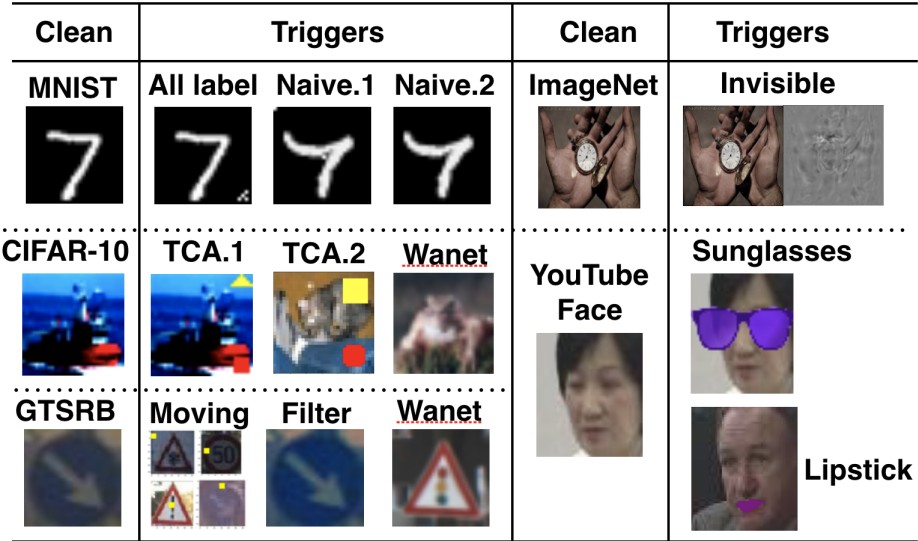

Figure 5: Pictures under "Triggers" are poisoned samples regarding different backdoored attacks. Pictures under "Clean" are clean samples for each dataset.

## A  DETAILS FOR NOVELTY DETECTION

The method in Table 4 is in the same order as shown in Fig. 3.

Table 4: Means and standard deviations of AUROC (%) for different methods on 100 UCI datasets.

| Ours | L1-Ball | K-Center | Parzen | Gaussian | K-Mean |
|---|---|---|---|---|---|
| $79 \pm 15.6$ | $65.4 \pm 25.4$ | $72.2 \pm 17.6$ | $77.6 \pm 18.6$ | $81.1 \pm 16.6$ | $76.3 \pm 17.9$ |
| 1-Nearest Neighbor | K-Nearest Neighbor | Auto-Encoder Network | Linear Programming | Principal Component | Lof Range |
| $78.8 \pm 18.3$ | $79.2 \pm 18.3$ | $73.7 \pm 17.9$ | $77.5 \pm 18.9$ | $75.1 \pm 18.3$ | $56.5 \pm 30.8$ |
| Nearest Neighbor Distance | Minimum Spanning Tree | Minimum Covariance Determinant | Self Organizing Map | Support Vector Machine | Minimax Probability Machine |
| $77.5 \pm 16.6$ | $78.8 \pm 17.4$ | $80.4 \pm 17.1$ | $77.6 \pm 17.4$ | $69.4 \pm 24.6$ | $66.4 \pm 33.3$ |
| Mixture Gaussians | Local Outlier Factor | Naive Parzen | Local Correlation Integral | | |
| $78.7 \pm 18.1$ | $76.1 \pm 19.7$ | $78.4 \pm 16.2$ | $60.7 \pm 23.9$ | | |

## B  DETAILS FOR OUT-OF-DISTRIBUTION DETECTION

Table 5 is for CIFAR-10 case and Table 6 is for CIFAR-100 case.

## C  DETAILS FOR BACKDOOR DETECTION

Fig. 5 shows the used triggers and the corresponding clean samples. Table 7 shows the details for backdoor detection.

Table 5: Results for CIFAR-10 in-distribution case (higher number implies higher accuracy). **Bold-face** shows the best performing algorithm, whereas underline shows the second best algorithm.

| Out-of-Distribution Datasets | Method | TPR95 (%) | AUROC (%) | AUPR (%) |
|---|---|---|---|---|
| Texture | Ours | 58.41 | 90.97 | 85.64 |
| | MSP | 40.75 | 88.31 | 97.08 |
| | Mahalanobis | 62.38 | 94.46 | 98.75 |
| | Energy Score | 47.47 | 85.47 | 95.58 |
| | GEM | **72.61** | **94.59** | **98.79** |
| SVHN | Ours | **88.31** | **96.23** | 98.49 |
| | MSP | 52.41 | 92.11 | 98.32 |
| | Mahalanobis | 79.34 | 95.72 | **99.04** |
| | Energy Score | 64.20 | 91.05 | 97.66 |
| | GEM | 79 | 95.65 | 99.01 |
| LSUN-Crop | Ours | 79.45 | 94.33 | 94.60 |
| | MSP | 69.07 | 95.64 | 99.13 |
| | Mahalanobis | 30.06 | 86.15 | 97.05 |
| | Energy Score | **91.89** | **98.40** | **99.67** |
| | GEM | 30.20 | 86.09 | 97.03 |
| LSUN-Resize | Ours | 69.92 | 92.03 | 92.54 |
| | MSP | 47.45 | 91.30 | 98.11 |
| | Mahalanobis | 35.64 | 88.12 | 97.45 |
| | Energy Score | **71.75** | **94.12** | **98.64** |
| | GEM | 35.45 | 88.09 | 97.43 |
| iSUN | Ours | 65.60 | 90.25 | 90.33 |
| | MSP | 43.40 | 89.72 | 97.72 |
| | Mahalanobis | 26.77 | 87.87 | 97.33 |
| | Energy Score | **66.27** | **92.56** | **98.25** |
| | GEM | 36.80 | 87.85 | 93.33 |
| Average Performance | Ours | **72.33** | **92.76** | 92.32 |
| | MSP | 50.63 | 91.46 | **98.07** |
| | Mahalanobis | 46.83 | 90.46 | 97.92 |
| | Energy Score | 68.31 | 92.32 | 97.96 |
| | GEM | 50.81 | 90.45 | 97.91 |

Table 6: Results for CIFAR-100 in-distribution case (higher number implies higher accuracy). **Boldface** shows the best performing algorithm, whereas underline shows the second best algorithm.

| Out-of-Distribution Datasets | Method | TPR95 (%) | AUROC (%) | AUPR (%) |
|---|---|---|---|---|
| Texture | Ours | 46.13 | 87.52 | 80.57 |
| | MSP | 16.71 | 73.58 | 93.02 |
| | Mahalanobis | **57.62** | 90.14 | 97.62 |
| | Energy Score | 20.38 | 76.46 | 93.68 |
| | GEM | 57.40 | **90.17** | **97.63** |
| SVHN | Ours | **90.69** | **96.41** | **98.58** |
| | MSP | 15.66 | 71.37 | 92.89 |
| | Mahalanobis | 51.35 | 89.25 | 97.52 |
| | Energy Score | 14.59 | 74.10 | 93.65 |
| | GEM | 51.51 | 89.40 | 97.57 |
| LSUN-Crop | Ours | 27.93 | 80.09 | 79.32 |
| | MSP | 33.44 | 83.71 | 96.32 |
| | Mahalanobis | 1.53 | 58.48 | 89.73 |
| | Energy Score | **64.01** | **93.41** | **98.59** |
| | GEM | 1.70 | 58.42 | 89.70 |
| LSUN-Resize | Ours | 39.80 | 88.15 | 86.79 |
| | MSP | 16.54 | 75.32 | 94.03 |
| | Mahalanobis | **67.20** | 93.97 | **98.70** |
| | Energy Score | 21.38 | 79.29 | 94.97 |
| | GEM | 67.09 | **94.01** | **98.70** |
| iSUN | Ours | 35.87 | 86.01 | 83.37 |
| | MSP | 17.02 | 75.87 | 94.20 |
| | Mahalanobis | 64.07 | 92.69 | **98.32** |
| | Energy Score | 19.20 | 78.98 | 94.90 |
| | GEM | **64.10** | **92.73** | **98.32** |
| Average Performance | Ours | 48.08 | **87.63** | 85.72 |
| | MSP | 19.87 | 75.97 | 94.09 |
| | Mahalanobis | 48.35 | 84.90 | 96.37 |
| | Energy Score | 27.91 | 80.44 | 95.15 |
| | GEM | **48.36** | 84.94 | **96.38** |

Table 7: Comparison results for backdoor detection (higher number implies higher accuracy).

| Datasets | Trigger | Metrics (%) | Ours | STRIP | Mahalanobis | GEM | MSP |
|---|---|---|---|---|---|---|---|
| MNIST | All label | TPR95 | 83.05 | 2.58 | 50.83 | **100** | 100 |
| | | AUROC | **96.13** | 44.69 | 90.78 | 50.43 | 50 |
| | | AUPR | **94.20** | 35.47 | 86.71 | 70.94 | 70.83 |
| MNIST | Naive.1 | TPR95 | **100** | 98.85 | 99.86 | 100 | 5.11 |
| | | AUROC | **97.50** | 97.32 | 97.49 | 53.95 | 51.64 |
| | | AUPR | 96.17 | 95.95 | **96.38** | 74.74 | 50.41 |
| MNIST | Naive.2 | TPR95 | 96.53 | 67.46 | 35.16 | **100** | 14.69 |
| | | AUROC | **97.28** | 93.67 | 78.63 | 53.51 | 58.14 |
| | | AUPR | **95.75** | 89.85 | 78.65 | 74.62 | 64.16 |
| CIFAR-10 | TCA.1 | TPR95 | **100** | 35.68 | 100 | 100 | 4.38 |
| | | AUROC | **97.50** | 83.00 | 97.49 | 50 | 49.23 |
| | | AUPR | 95.47 | 73.22 | **97.84** | 76.32 | 52.64 |
| CIFAR-10 | TCA.2 | TPR95 | **100** | 27.86 | 100 | 100 | 0.02 |
| | | AUROC | **97.50** | 68.79 | 97.49 | 50 | 29.90 |
| | | AUPR | **97.63** | 72.41 | 95.86 | 67.86 | 18.05 |
| CIFAR-10 | Wanet | TPR95 | 37.87 | 0.07 | 20.35 | 22.90 | **100** |
| | | AUROC | **92.74** | 34.97 | 50.61 | 57.81 | 50 |
| | | AUPR | **89.95** | 37.42 | 57.30 | 68.48 | 74.87 |
| GTSRB | Moving | TPR95 | **99.99** | 54 | | | |
| | | AUROC | **85.39** | 7.29 | Fail: dependent data dimensions | | |
| | | AUPR | **96.96** | 89.07 | | | |
| GTSRB | Filter | TPR95 | **85.39** | 7.29 | | | |
| | | AUROC | **96.54** | 38.92 | Fail: dependent data dimensions | | |
| | | AUPR | **95.42** | 38.81 | | | |
| GTSRB | Wanet | TPR95 | **100** | 1.24 | 0.51 | 100 | 100 |
| | | AUROC | **97.50** | 36.31 | 54.46 | 50 | 50 |
| | | AUPR | **97.62** | 39.53 | 48.92 | 75.23 | 75.23 |
| YouTube Face | Sunglasses | TPR95 | 73.37 | 83.03 | 71.64 | **98.58** | 13.06 |
| | | AUROC | **95.21** | 94.80 | 94.38 | 84.29 | 66.55 |
| | | AUPR | 93.00 | **95.54** | 94.63 | 88.83 | 53.27 |
| YouTube Face | Lipstick | TPR95 | **96.64** | 90.14 | 90.88 | 94.18 | 3.73 |
| | | AUROC | **97.21** | 93.15 | 93.26 | 80.80 | 50.14 |
| | | AUPR | **96.30** | 94.98 | 95.09 | 86.53 | 53.27 |
| sub-ImageNet | Invisible | TPR95 | **100** | 7.01 | 0.5 | 100 | 51.40 |
| | | AUROC | **97.49** | 66.26 | 4.78 | 50 | 93.61 |
| | | AUPR | **96.53** | 62.83 | 12.27 | 75.26 | 92.46 |
| Average Performance | | TPR95 | 89.40 | 39.60 | 56.97 | **91.57** | 39.24 |
| | | AUROC | **96.68** | 70.30 | 75.94 | 58.08 | 54.92 |
| | | AUPR | **95.42** | 68.76 | 76.37 | 75.88 | 60.52 |

