# OpenReview forum: "An Upper Bound for the Distribution Overlap Index and Its Applications"
_ICLR.cc/2023/Conference — Submitted to ICLR 2023_

### Official Review · Reviewer_yR91 · 2022-10-23

**Confidence:** 4
**Correctness:** 3
**Technical Novelty And Significance:** 3
**Empirical Novelty And Significance:** 3
**Recommendation:** 6

**Clarity, Quality, Novelty And Reproducibility:**

In addition to several weaknesses highlighted above, it would be useful for a reader for reproduciblity and clarity if the following points could be clearly discussed in the paper:

- More discussion should be added on choices of the conditional function g. Currently, authors use indication function $I(||x|| < r)$. Can authors present some motivation behind this choice and other choices that can be worth exploring in the future?
- In the abstract authors mention that their one-way classifier "requires only a small number of in-class samples to be accurate." However, this argument is not substantiated with any theoretical or concrete empirical justification. In my opinion, this statement should be toned down to precisely reflect the experimental finding. Moreover, in general, empirical experiments can not show that the proposed bound requires only a small number of samples to be accurate.
- More details about the baseline methods compared in Section 4.4 should be included in the paper. Are there hyperparameter choices in the baseline methods that impact the final results? If yes, how did the authors choose those hyperparameters in their comparison?
- For high-dimensional benchmark datasets, authors use WideResNets to extract features that I assume to act as the input $x$. Can authors provide more details on how this Resnet is trained? Moreover, the experimental setup for Section 4.4.2 is missing. More details should be added to justify how the comparisons made in Tables 2 and 3 are fair with other baseline methods.

**Other Writing suggestion**:
- Language in Theorem 2 should be significantly improved. In its current form, it is very confusing. While it is not hard to infer the real meaning using parenthesis like in "If a model is trained on D with p accuracy on D (including the overlap area between D and D∗)" doesn't make it clear that p is the accuracy on D which is restricted to the overlap region. It would be great if authors can formally define these quantities. Moreover, the last phrase "is upper bounded due to (14)" is unclear. What is this quantity upper bound on? Is it the overlap index?
- (Very minor) For Definitions 1, 2, and 3, authors may consider defining P and Q distribution in the text before these definitions to avoid redundancy


I believe that all the suggestions made above can be taken into account during the rebuttal phase, and I look forward to the author's response and will be happy to change my score if these questions are answered satisfactorily.

**Strength And Weaknesses:**

**Strengths**
-  The paper presents an interesting yet simple upper bound of the overlap index. The upper bound is distribution-free and can be computed only with samples.
-  An interesting application of the bound on one-class classification is presented in the paper.  Experimental evidence for novelty detection, out-of-distribution detection, and backdoor detection shows the efficacy of the method. Moreover, the proposed one-way classifier does not need to train any parameters.



**Weaknesses**
- In Theorem 2, authors present an approximation to the accuracy of a model under distribution shift. However, it is unclear if this bound is practically useful. Moreover, there are no experiments highlighting the efficacy or practicality of the bound. Specifically, this may happen because we can not compute the term q in the bound.
- Abstract and introduction of the paper should be toned down with respect to the above contribution. In particular, it should be made clear that the application to accuracy estimation is only of theoretical interest.
- Several crucial experimental details and design choices are missing in the paper. Please refer to the response below for details.


**Summary Of The Paper:**

The paper proposed an upper bound to overlap index to measure the distributional distance between two distributions in a distribution-free setting, i.e. when we are only given access to finite samples from the underlying distributions. The authors first present a theoretical result to approximate the accuracy of a model with their upper bound on the overlap index.  Second, the authors employ this bound to study the distribution membership of given examples. Experimental results on out-of-distribution and novelty detection show that the proposed bound outperform existing heuristics on average in the AUROC metric.

**Summary Of The Review:**

Overall, the paper presents an interesting yet simple upper bound of the overlap index. The upper bound is distribution-free and can be computed only with samples. An interesting application for one-class classification is discussed in the paper. However, the section with the application of accuracy estimation under distribution shift is a bit skim, and currently, it is unclear if the proposed upper bound on the overlap index can be leveraged in any non-trivial manner. Moreover, the paper lacks some crucial experimental and baseline details.

---

> ### Author Response · Authors · 2022-11-16
> **Corresponding changes and responses**
>
> We thank you for taking the time to evaluate our work and appreciate the points you have raised. Please see our responses below:
>
> Responses to weakness 1 & 2:
> Due to the page limit, we had not included details on experiments with the distribution shift. However, per your concern, we have added a set of results (in Figure 2) and a discussion on the distribution shift in the updated paper (in Section 3). The newly added results consider the validation of Theorem 2 in backdoor attack scenarios [1] by considering the clean data distribution as D and the contaminated data distribution as D^*. The experimental results show consistency with Theorem 2. We also mentioned the validation (in Introduction).
>
> Responses to clarity 1:
> With the indicator function I(||x||<r), one can directly apply this “r” to corollary 1, saving computations. Additionally, it is the most simple way to separate a space nonlinearly. However, we believe other RBF kernel-based indicator functions are worth exploring and will be considered in future work. We have further discussed the motivation for the choice of indicator function and consideration of RBF kernels-based indicator functions (in Remark 3).  Please also read the new two paragraphs related to improvements in Section 4.4.2 for information on how the choice of indicator functions will affect the performance of our approaches.
>
> Responses to clarity 2:
> We have reworded and clarified the corresponding statement (in our Abstract and the third paragraph in the Introduction).
>
> Responses to clarity 3:
> We downloaded all the baseline algorithms directly from the websites provided by their authors as well as the related datasets and models. Therefore, we consider the hyperparameters to be well-chosen for those baseline works on the related datasets. The only exception is backdoor detection, in which we created our own backdoor models. However, we have carefully fine-tuned the baselines’ hyperparameters so that they show the best performance compared with other hyperparameter choices. We added an “overall setup” at the beginning of Section 4.4 to clarify the hyperparameter choice.
>
> Response to clarity 4: The ResNet models (well-trained on CIFAR-10 and CIFAR-100 training datasets) are directly downloaded from the work GEM [2]’s website, which also provides the corresponding feature extraction functions. We evaluated all the methods with optimal hyperparameters in the same feature spaces using the provided ResNet models and feature extractors. Therefore, the comparison is fair. We have added these details in the first paragraph of Section 4.4.2.
>
> Responses to writing suggestion 1:
> Yes, (14) is on the overlap index. But thank you for pointing out the sentences that could be confusing. To address your concerns, we have rewritten Theorem 2 to avoid any confusion.
>
> Responses to writing suggestion 2:
> We appreciate your input. We have made the corresponding change in Section 2.1.
>
> All the changes are in blue color in the updated manuscript.
>
> [1] Gu, Tianyu, et al. "Badnets: Evaluating backdooring attacks on deep neural networks." IEEE Access 7 (2019): 47230-47244.
>
> [2] Morteza, Peyman, and Yixuan Li. "Provable guarantees for understanding out-of-distribution detection." Proceedings of the AAAI Conference on Artificial Intelligence. Vol. 8. 2022.

---

> > ### Comment · Reviewer_yR91 · 2022-11-19
> > **Reply**
> >
> > I thank the authors for their effort in rebuttal and detailed response. I am unclear about Figure 2 in the sense that how is upper bound on accuracy is useful if empirically they are large than accuracy. Could the authors please elaborate?

---

> > > ### Author Response · Authors · 2022-11-20
> > > **Elaboration of usefulness in Figure 2**
> > >
> > > Figure 2 validates Theorem 2 by showing a larger upper limit than the actual model accuracy. Moreover, Figure 2 reveals how Theorem 2 is useful in practice:
> > >
> > > Point 1. The difference between actual model accuracy and our upper bound accuracy provides an indication as to how large the domain shift is. From Figure 2, the difference decreases as the domain shift reduces. When the domain shift vanishes, the actual model accuracy and our upper bound accuracy meet.
> > >
> > > Point 2. From Figure 2, the estimated upper bound on accuracy in Theorem 2 varies with the choice of norms and feature spaces. We can use this variation of Theorem 2 to infer useful information from the data. Specifically, the inference is that after trying multiple candidate norms, a low estimated upper bound on accuracy in Theorem 2 in a particular feature space implies a high likelihood of distinguishing clean and poisoned data distributions in that particular feature space. For example, in Figure 2 MNIST and YouTube, the input space with L_infinity norm gives the lowest estimated upper bound on accuracy. Therefore, the inference is that the clean and poisoned data distributions are likely to be distinguished in the raw image. The inference is supported by the fact that the MNIST used the “All label” trigger and YouTube used the “Sunglasses” trigger as shown in Figure 5 in the appendix, where even vision inspection can easily distinguish between clean and poisoned samples. As for GTSRB and ImageNet, the input space has the highest upper limit lines. Therefore, the inference is that the clean and poisoned data distributions are less likely to be distinguished in the raw data space. The inference is also supported by the fact that GTSRB uses the “Filter” trigger and ImageNet uses the “Invisible” trigger as shown in Figure 5, where the vision inspection barely discriminates between clean and poisoned samples. Moreover, the hidden layer space gives the lowest estimated upper bound on accuracy for GTSRB and ImageNet. Therefore, one may try the hidden layer space to distinguish the clean and poisoned data samples rather than other spaces.
> > >
> > > Practical scenario: Consider that a user is testing a backdoored network on a contaminated dataset and assume that the user has a small clean validation dataset (thus can approximate the clean data distribution). Through Point 1, the user knows that a domain shift exists in the testing dataset. With Point 2, the user knows the best feature space and norm to most likely find and remove those poisoned samples to purify the contaminated dataset.
> > >
> > > We will further clarify the above points in the final camera-ready version of our paper.

---

> > > > ### Comment · Reviewer_yR91 · 2022-11-28
> > > > **Thanks for your reply**
> > > >
> > > > I thank the authors for clarification. I agree that the upper bound on accuracy will be useful to detect distribution shifts. However, distribution detection can also be accomplished with two-sample testing (e.g., training a discriminator on datasets A and B and checking its performance on holdout data from A and B). In light of these points, I stay with my score of 6.

---

### Official Review · Reviewer_S8uA · 2022-10-25

**Confidence:** 3
**Clarity, Quality, Novelty And Reproducibility:** The paper is well written and well st…
**Correctness:** 3
**Technical Novelty And Significance:** 3
**Empirical Novelty And Significance:** 2
**Recommendation:** 5

**Strength And Weaknesses:**

Strength:
Interesting idea
Weakness:
Minor improvement in results AUPR is not better than others or not that much increased.

**Summary Of The Paper:**

The authors define an upper bound for the overlap index and then by converting it into confidence score they can train a one-class classifier. Since the bound is estimated on unknown distribution then the classifier is also distribution free.

**Summary Of The Review:**

Given the comments, I vote for borderline reject.

---

> ### Author Response · Authors · 2022-11-16
> **Corresponding changes and responses**
>
> We thank you for taking the time to evaluate our work and appreciate the points you have raised. Please see our responses below:
>
> 1. This paper’s focus is on developing a new general approach for one-class classification problems based on an efficient computation of estimates of the overlap index between two empirical probability distributions, which is fundamentally a new approach and has general applicability. Specifically, the novelty of this paper is that we provide a new approach to compute a distribution-free upper bound for the overlap index, prove its correctness theoretically, and show its practical utility in one-class classification. We would like to highlight that our approach provides the highest performance in terms of TPR95 and AUROC metrics. We also measured the memory and computation costs in Table 1 in the updated paper. Our approach uses the least memory and is one of the fastest approaches.
>
> 2. The AUPR is relatively a little lower (but overall in a similar range) compared to other methods. However, the overall performance of our approach can be improved by using more indicator functions and an iterative approach. We conducted related experiments and show the improvement in Figure 4 and Table 2 in the updated paper. We also noticed that the current out-of-distribution datasets are unbalanced. Therefore, we also tested all the methods on balanced datasets including MNIST, CIFAR-10, GTSRB, YouTube Face, and ImageNet using the backdoor settings in the paper. The results are provided in the updated paper in Table 3. The highlight is that in these balanced datasets, our approach has the highest AUROC and AUPR. Furthermore, it is to be noted that the numerical results of the methods can be potentially further increased by tuning the choice of our condition functions and using more iterations, which we discussed in Section 4.4.2.
>
> 3. We updated our paper by introducing more metrics (i.e., timing and memory in Table 1), improving the overall performance of our approach using more indicator functions and iteration (in Figure 4 and Table 2), and testing all the methods on balanced datasets (in Table 3) to strengthen that our approach is valid and outperforms the state-of-the-art methods. The corresponding discussions in the text are also added (in Section 4.4.2). We moved old tables to Appendix for novelty detection, out-of-distribution detection, and backdoor detection.
>
> 4. Our approach originally used ten indicator functions for novelty detection. However, we have evaluated our approach with 50 indicator functions being used. The mean AUROC becomes 81.9 (i.e., the highest), and the standard deviation becomes 13.8 (i.e., the lowest). Therefore, our approach outperforms all the baseline methods. We will further update Figure 3 and Table 4 in the final camera-ready version of our paper.
>
> The changes are in blue color in the updated manuscript.

---

### Official Review · Reviewer_NVdx · 2022-11-02

**Confidence:** 2
**Correctness:** 3
**Technical Novelty And Significance:** 2
**Empirical Novelty And Significance:** 2
**Recommendation:** 5

**Clarity, Quality, Novelty And Reproducibility:**

At a high level, the algorithm for Compute Bound is very similar to other methods of distribution distance which uses statistics of distribution to compute the distance between them. I would suggest the authors compare it theoretically with other methods, where it can work, and where it will fail.

While the authors talk about how their method can be useful for distribution shifts, there are no experiments in the paper regarding that. How shall we validate that Theorem 2 actually holds in practice?

When we say the distance between two distributions, isn't it supposed to be at the input space itself? when we compute the distance on the extracted feature space, does that really allow for a good distance metric? because then the distance is dependent on the model.


**Strength And Weaknesses:**

Strengths
- The paper is nicely written

Weakness
- The results on novelty class detection are not convincing. Traditional methods perform almost similarly to the proposed method
- The performance for out-of-distribution detection is also not convincing. There is no clear winner, and the proposed method is only good on half of the datasets. Similar comment for Table 3.
- The authors do not compare with Deep One-Class Classification, which is a relevant work.
- Figure 3 is not well presented. The captions should be self-explanatory.
- Overall it is not clear what is the utility of this work compared to other works in the literature, as it hardly introduces a new concept, findings, or better performance.





**Summary Of The Paper:**

The paper proposes a method to compute the distance between two distributions.
The authors show how this can be used for novelty detection, out-of-distribution classification, and the like. The authors present results primarily focusing on these tasks, and compare them with traditional novelty detection methods.

**Summary Of The Review:**

The paper tries to attack an important problem of measuring the distance between two distributions. However, the technique that the paper proposes is quite similar to the statistics-based method for distribution distance computation. The results section of the paper is missing a comparison with some commonly used techniques, and the proposed method does not stand out as the clear winner in any of the settings.

---

> ### Author Response · Authors · 2022-11-16
> **The corresponding changes and responses to points in clarity**
>
> Response to weakness 5 & clarity 1:
> Distribution-free and model-independent are the paper’s key novelty. To illustrate the usefulness of this novel approach, consider the comparison with Gaussian-based methods. If there exists dependence between data dimensions, the Gaussian-based methods will fail because the estimated covariance matrices will not be full rank, leading to an operation error when calculating the inverse of the estimated covariance matrices. Indeed, we empirically observed this operation error when using the Gaussian-based methods for backdoor detection. In contrast, our approach works since it does not require any distribution models or finding the inverse of any matrices. We have added this point (in the last paragraph of Section 4.4.2). As for other distribution-free baseline methods, our approach will empirically outperform them, which is supported by the new results in novelty detection (i.e., our approach has the highest average AUROC and lowest Std. of AUROC).
>
> Response to clarity 2:
> Due to the page limit, we had not included details on experiments with the distribution shift. However, per your concern, we have added a set of results (in Figure 2) and a discussion on the distribution shift in the updated paper (in Section 3). The newly added results consider the validation of Theorem 2 in backdoor attack scenarios [1] by considering the clean data distribution as D and the contaminated data distribution as D^*. The experimental results show consistency with Theorem 2. We also mentioned the validation (in Introduction).
>
> Response to clarity 3: Calculating the distribution distance can be in any space. Indeed, all the baseline methods for out-of-distribution detection and backdoor detection in our paper calculated their distribution distances in feature spaces in their original works. It is true that different models will generate different feature vectors for the same distribution. However, the comparison is considered fair and convincing as long as all the approaches calculate their distribution distances in the same feature spaces, which we did. We have added the point (in the first paragraph in Section 4.4.2).
>
> All the changes are in blue color in the updated manuscript.
>
> [1] Gu, Tianyu, et al. "Badnets: Evaluating backdooring attacks on deep neural networks." IEEE Access 7 (2019): 47230-47244.

---

> ### Author Response · Authors · 2022-11-16
> **Corresponding changes and responses to points in weakness**
>
> Thank you for taking the time to evaluate our work and appreciate the points you have raised. Please see our responses below:
>
> Supplementary points to your summary of our paper:
> The key novelty of our paper is that it proposes a distribution-free and model-independent upper bound for the overlap index and theoretically proves its correctness. Additionally, we compare our approach with both traditional methods and the most recent deep network-based approaches from 2018 to 2022.
>
> Response to weakness 1 & 2:
> 1. This paper’s focus is on developing a new general approach for one-class classification problems based on an efficient computation of estimates of the overlap index between two empirical probability distributions, which is fundamentally a new approach and has general applicability. Specifically, the novelty of this paper is that we provide a new approach to compute a distribution-free upper bound for the overlap index, prove its correctness theoretically, and show its practical utility in one-class classification. We would like to highlight that our approach provides the highest performance in terms of TPR95 and AUROC metrics. We also measured the memory and computation costs in Table 1 in the updated paper. Our approach uses the least memory and is one of the fastest approaches. Therefore, we would like to say that our experimental results are sufficient and convincing to support the claim that our approach is comparable to state-of-the-art methods and valid for novelty detection, out-of-distribution detection, and backdoor detection.
> 2. The AUPR is relatively a little lower (but overall in a similar range) compared to other methods. However, the overall performance of our approach can be improved by using more indicator functions and an iterative approach. We conducted related experiments and show the improvement in Figure 4 and Table 2 in the updated paper. We also noticed that the current out-of-distribution datasets are unbalanced. Therefore, we have tested all the methods on balanced datasets including MNIST, CIFAR-10, GTSRB, YouTube Face, and ImageNet using the backdoor settings in the paper. The results are provided in Table 3 in the updated paper. The highlight is that in these balanced datasets, our approach has the highest AUROC and AUPR. Additionally, it is to be noted that the numerical results of the methods can be potentially further increased by tuning the choice of our condition functions and using more iterations, which we discussed in Section 4.4.2.
> 3. We updated our paper by introducing more metrics (i.e., timing and memory in Table 1),  improving the overall performance of our approach using more indicator functions and an iterative approach (in Figure 4 and Table 2), and testing all the methods on balanced datasets (in Table 3) to strengthen that our approach is valid and outperforms the state-of-the-art methods. The corresponding discussions in the text are also added (in Section 4.4.2). We moved old tables to Appendix for novelty detection, out-of-distribution detection, and backdoor detection.
> 4. Our approach originally used ten indicator functions for novelty detection. However, we have evaluated our approach with 50 indicator functions being used. The mean AUROC becomes 81.9 (i.e., the highest), and the standard deviation becomes 13.8 (i.e., the lowest). Therefore, our approach outperforms all the baseline methods. We will further update Figure 3 and Table 4 in the final camera-ready version of our paper.
>
> Response to weakness 3:
> Deep One-Class Classification is relevant to our work since they both address one-class classification problems. However, they are inherently different and cannot be compared experimentally because Deep One-Class Classification is designed for anomaly detection whose training data has anomaly samples, whereas our work considers novelty detection, out-of-distribution detection, and backdoor detection, which do not allow the training data to contain anomaly samples.
>
> Response to weakness 4:
> Thanks for pointing out the issue. We have added the necessary information in the caption to ensure that it is self-explanatory (in Figure 5).
>
> Response to weakness 5: We derived a new upper bound for the overlap index. Additionally, we applied the upper bound to one-class classification problems and showed a better performance of our approach than other works in the literature. Specifically, for novelty detection, the new results (Average AUROC is 81.9 and Std. of AUROC is 13.8) show that our approach has the best performance compared to other baseline methods. For out-of-distribution detection, the new results show that our approach has a much higher TPR95 and AUROC than the baseline methods in Table 2 and Figure 4, and the AUPR is on the same level. Additionally, the baseline methods are Gaussian-based, which may fail in some scenarios. Please read the next comment block for more details.
>
> All the changes are in blue color in the updated manuscript.

---

> ### Comment · Reviewer_NVdx · 2022-12-01
> **Thanks for the reply**
>
> I thank the authors for the reply, and really appreciate the additional experiments.
>
> Just to clarify, deep one class classification do not need any other sample, other than a single class and/or distribution, which is also the case for this problem in hand. Moreover, the fact that the bound is dependent on the model features makes the claim of "model-independent upper bound for the overlap index" vacuous.
>
> There are concerns which I still have about the work, and I still think that the work is not well evaluated (against more sophisticated baselines), does not add much value on top of the existing literature.
>
> My rating thus remains unchanged.

---

> > ### Author Response · Authors · 2022-12-02
> > **Response**
> >
> > Thank you for your feedback. We have implemented our approach on MNIST and CIFAR-10 for the anomaly detection considered in the Deep One-Class Classification paper. The average AUROC for our approach is 95.2 on MNIST and 67.7 on CIFAR-10, whereas Deep One-Class Classification only has 94.8 on MNIST and 64.81 on CIFAR-10. Additionally, we noticed that Deep One-Class Classification needs a large training dataset (i.e., 6000 for MNIST and 5000 for CIFAR-10) to reach the above performance, whereas our approach requires only a small dataset (i.e., 500 for MNIST and 100 for CIFAR-10). If Deep One-Class Classification uses the same small datasets as ours, its AUROC becomes 91.5 for MNIST and 61.1 for CIFAR-10. Therefore, our approach outperforms Deep One-Class Classification. Our approach is implemented on the raw image space for a fair comparison.
> >
> > The “model-independent” in our response means “distribution-model independent”, which is same to “distribution-free".  Sorry for the confusion. However, the word “model-independent” appears nowhere in our paper. Therefore, our manuscript (both initial version and updated version) is very clear at this point.

---

### Decision · Program_Chairs · 2023-01-20

**Decision:**

Reject

**Justification For Why Not Higher Score:**

The 3 reviewers agreed on a consensus of weak reject, there are still some elements that have to be justified.

**Justification For Why Not Lower Score:**

N/A

**Metareview: Summary, Strengths And Weaknesses:**

The paper proposes to consider a distance between distributions by focusing on the overlap index between the two distributions. It proposes a method to compute an upper bound on this index which is distribution-free. The bound then converted into a confidence score function which is efficient to compute.

Strengths:
-Method simple and efficient
-interesting applications in 1-class classification, novelty/anomaly detection

Weaknesses:
-Experimental evaluation of the work is not sufficient are requires additional study
-Position of the work with respect to other methods of the literature is unclear and requires better justification.
-Global writing can be improved.

After the rebuttal phase, authors have provided answers and precisions to the remarks of the reviewers. This has been appreciated and taken into consideration. During discussion, reviewers agreed that the presentation of work still requires some work (discussion remains superficial) and the significance with respect to other methods of the literature has to be better justified both methodologically and experimentally. Despite the interestingness of the paper, there was a consensus on a weak reject.

Overall the paper has an interesting potential which is not justified in a convincing manner yet. I propose then rejection.